

# Use of *Callistemon citrinus* as a gastroprotective and anti-inflammatory agent on indomethacin-induced gastric ulcers in obese rats

Jonathan Saúl Piñón-Simental[1], Luis Alberto Ayala-Ruiz[1], Luis Gerardo Ortega-Pérez[1], Oliver Rafid Magaña-Rodríguez[1], Esperanza Meléndez-Herrera[2], Asdrubal Aguilera-Méndez[3] and Patricia Rios-Chavez[1]

[1] Biologia, Universidad Michoacana de San Nicolás de Hidalgo, Morelia, Michoacan, Mexico
[2] Instituto de Investigaciones sobre los Recursos Naturales, Universidad Michoacana de San Nicolás de Hidalgo, Morelia, Michoacan, Mexico
[3] Instituto de Investigaciones Químico Biológicas, Universidad Michoacana de San Nicolás de Hidalgo, Morelia, Michoacan, Mexico

## ABSTRACT

**Background:** Obesity leads to an elevated risk of developing gastrointestinal disease such as gastric ulcers. *Callistemon citrinus* leaf extract has shown antioxidant, antimicrobial, hepatoprotective, and chemoprotective effects against colon cancer. The aim of this study is to evaluate the gastroprotective effect of *C. citrinus* leaf extract on indomethacin-induced gastric ulcers in obese rats.

**Methods:** Gastric ulcers were induced in female obese Wistar rats using a single oral dose of indomethacin (IND). In the first stage, the rats were fed with a high fat sugar diet (HFSD) for 15 weeks to induce obesity and, at the same time, the diet of the other group of animals included daily administration of ethanolic *C. citrinus* leaf extract (250 mg/kg) in addition to HFSD. In the second stage, gastric ulcers were induced with IND (30 mg/kg). The gastroprotective activity of *C. citrinus*, the inflammatory enzyme activities, and cytokines in the stomach were determined.

**Results:** *C. citrinus* produced a reduction of gastric lesions caused by IND. Myeloperoxidase (MPO), cyclooxygenase-2 (COX-2), and 5-lipoxygenase (5-LOX) activities also decreased. Although inflammatory biomarkers such as TNFα, IL-6, AOPP, and leptin were significantly decreased by *C. citrinus*, adiponectin levels increased. Moreover, *C. citrinus* decreased weight gain and morphological and biochemical parameters.

**Conclusion:** The use of indomethacin in rats fed with a high fat-sugar diet increased gastric ulcers. Gastroprotective effect of *C. citrinus* in obese rats is attributed to the reduction of pro-inflammatory cytokines and the inflammatory enzymes.

Corresponding author
Patricia Rios-Chavez,
prchavez17@gmail.com

## INTRODUCTION

Indomethacin (IND) is a non-steroidal anti-inflammatory drug (NSAID) used in the treatment of diseases involving pain, fever, and muscle skeletal disorders, and in the relief of osteoarthritis, rheumatoid arthritis, and other comorbidities (*Ghosh, Alajbegovic & Gomes, 2015*). Common adverse effects are dyspepsia, gastroduodenal ulcer, gastrointestinal (GI) bleeding, perforation, edema, myocardial infarction, stroke, and a reduced glomerular filtration rate (*Wongrakpanich et al., 2018*). IND belongs to the group of semi-selective NSAIDs and has a high affinity for cyclooxygenase-2 (COX-2). However, it also inhibits cyclooxygenase-1 (COX-1). COX-1 is a constitutive isoform known as the housekeeping enzyme because of its role in maintaining physiological functions such as GI cytoprotecting, vascular smooth muscle tone modulation, regulation of renal water, sodium metabolism, and platelet aggregation (*Biava, 2018*). Conversely, COX-2 is an upregulated isoform presented during an inflammation process. The widespread use of NSAIDs (diclofenac, ibuprofen, IND, and celecoxib) causes gastric and intestinal mucosa damage, as well as injury and malfunction of the organs involved in the absorption and assimilation of food (*Maseda & Ricciotti, 2020*). *Cheng, Lu & Yen (2017)* reported that gastric ulcers have various processes such as the activation of phospholipase A, COX-2, myeloperoxidase (MPO), and pro-inflammatory cytokines (TNF-α, IL-6, and leptin). Moreover, they found a decrease of COX-1 activity and adiponectin level.

Peptic ulcer disease (PUD) includes gastric ulcers (in the stomach) and duodenal ulcers (in the proximal duodenum). The factors involved in the development of peptic ulcers are *Helicobacter pylori* infection, increase of gastric acid, and use of NSAIDs. *Boylan et al. (2014)* reported that gastric ulcers were more frequently found in obese men with a body mass index (BMI) greater than 30.0 kg/m$^2$ compared to men with a BMI of 23.0 kg/m$^2$. Additionally, 62% of the obese persons with gastric ulcers reported the regular intake of aspirin or NSAIDs. *Kim, Kim & Lee (2017)* showed the association of age, BMI, obesity and low intake of fiber, calcium, and sodium with peptic ulcers in Korean women, but not in Korean men. *Zhao et al. (2017)* found that a high fat diet induced obesity and accelerated the formation of gastric lymphoid follicles in the presence of *Helicobacter suis*, *via* the activation of nuclear factor kappa B (NF-κB) signaling and lymphoid chemokines. Another study using a Mendelian randomization showed that abdominal obesity had a high correlation with peptic ulcer disease (*Li, Chen & Chen, 2022*). *Ren et al. (2022)* found a relationship between BMI and peptic ulcer in Wuwei women, but not in men. However, *Périco et al. (2019)*, found that men were more prone to gastric ulcers than women, and the trend was reduced in elderly people.

*Nam (2017)*, showed that fat deposition is associated with gastrointestinal disease. *Emerenziani et al. (2020)* reported the link between obesity and gastric inflammatory diseases that was also found in children (*Tambucci et al., 2019*). In addition, *Yim, Kim & Lee (2021)* reported that age, body mass index, systolic blood pressure, educational level, even the number of household members are import factors to develop peptic ulcer disease.

*Hermansson et al. (2009)* reported an increase of NSAIDs consumption in the period of 1975 to 2002 in Sweden, especially in elderly women. Moreover, *Lind et al. (2017)* showed

the importance of developing studies based on sex when prescribing drugs. Indeed, women present more adverse drug events than men (*Rydberg et al., 2014*). *Farkouh et al. (2021)* reported that anti-inflammatory drugs can produce more damage in women than in men. Fat distribution is also different in men and women and obesity is higher in women than in men (*Ohara et al., 2023*). *Alhalabi (2023)* reported that men had a higher risk of peptic ulcer and complications, as bleeding, than women (2.4% and 1.2%, respectively). The opposite was reported by *Abed & Sameen (2023)*; however, women in this study had a chronic disease.

IND has the highest ulcerogenic potential in murine models and humans compared to other drugs (*Antonisamy et al., 2016*). In addition, it is used as a standard model when searching new compounds with pharmacological potential in the treatment of ulcers in the gastrointestinal tract (*Tamaddonfard et al., 2019*). Obesity is related to gastrointestinal complications such as gastroesophageal reflux disease, Barrett's esophagus, erosive gastritis, pectic ulcers, and neoplastic tumors (*Camilleri, Malhi & Acosta, 2017*). *Kim et al. (2007)* reported the relationship between obesity/overweight and the presence of gastric erosions or ulcers.

*Callistemon citrinus* (Myrtaceae) has been used in several countries' traditional medicine for the treatment of hemorrhoids, dysentery, rheumatism, tuberculosis, and bronchitis (*Laganà et al., 2020*). *C. citrinus* has regulatory activity on α-glucosidase (*Fayemi et al., 2019*) as well as antioxidant, cardioprotective, hepatoprotective, and chemoprotective activities (*López-Mejía et al., 2019*; *Kumar et al., 2020*; *López-Mejía et al., 2021*). Eucalyptin, blumenol, gallic acid, proto catechic acid, quercetin, catechin, astragalin, 6,7-dimethyl-5,7-dihydroxy-4′-methoxy flavone, sideroxylin, and syzalterin are the major phenolics and flavonoids found in *C. citrinus* (*Cuong et al., 2016*; *Khanh et al., 2016*). Flavonoids present with antioxidant, gastric ulcer-healing, anti-secretory, and cytoprotective pharmacological properties used in gastric problems (*Mota et al., 2009*).

*Petronilho et al. (2013)* used gas chromatography-mass spectrometry (GC-MS) and two-dimensional gas chromatography (GC x GC) to analyze the chemical composition of *C. citrinus*. The leaves have a high content of terpenes, and 1,8-cineole, limonene, and α-terpineol are the main components. *López-Mejía et al. (2021)* determined the concentration of some of the major phytochemicals in *C. citrinus* leaf ethanolic extract. Limonene has been reported to have gastroprotective effects on lesions induced by ethanol (*Moraes et al., 2009*; *de Souza et al., 2019*). 1,8-Cineole has antimicrobial, antioxidant, and anti-inflammatory effects (*Campos & Berteina-Raboin, 2022*), and α-terpineol has cardiovascular, anticancer, antioxidant, antinociceptive, and antiulcer activities (*Khaleel, Tabanca & Buchbauer, 2018*).

*Ortega-Pérez et al. (2022)* showed that the lethal dose ($LD_{50}$) of *C. citrinus* was higher than 4,000 mg/kg in an acute single dose toxicity test, and there were no toxic effects or significant changes in the body weight (BW) in the sub-chronic toxicity study. Moreover, the biochemical parameters aspartate aminotransferase (AST), alanine transaminase (ALT), lactate dehydrogenase (LDH), total protein, and albumin and the histological study were similar as the control group. Another study reported the safety of using *C. citrinus* leaf

extract for 22 weeks (*López-Mejía et al., 2019*). The results of these studies confirmed the safety of *C. citrinus* leaf extract.

Although *C. citrinus* has antioxidant and anti-inflammatory properties, its use as a possible gastroprotective mechanism has not yet been reported. The development of obesity in women is faster than in men, but most studies on high fat diets in rodents have been performed in male rats (*Maric et al., 2022*). *Fernandes et al. (2016)* reported that many diseases including obesity develop in a similar way in animal models as in humans.

It is unclear whether the high intake of fat and sugars in diet and the consumption of NSAID predispose the development of gastric ulcers. This study was designed to assess the effect of *C. citrinus* on oxidative and inflammatory process involved in IND-induced gastric ulcers in female rats fed with a high fat-sucrose diet. Our results show the gastroprotective activity of *C. citrinus* that attenuate gastric ulcers and prevent oxidative stress.

## MATERIALS AND METHODS

### Experimental design

The aim of this study was to determine the relationship between obesity and the predisposition of gastric ulcers when taking anti-inflammatory drugs, and the gastroprotective effect of ethanolic leaf extract of *C. citrinus* in Wistar rats.

The study was carried out in two stages. The first consisted of inducing obesity in female Wistar rats, while another group with the same hypercaloric diet was administrated ethanolic leaf extract of *C. citrinus* for 15 weeks. Once obesity was established, the second stage was comprised of inducing gastric ulcers with IND and then determining the gastroprotective effect of *C. citrinus*.

### Biological material and preparation of ethanolic extract

Leaves of *C. citrinus* were collected in Morelia, Mexico (19°41′11.3″N latitude and 101°12′18.4″O longitude). A voucher specimen of the plant was deposited at the Herbarium of the Faculty of Biology of the Universidad Michoacana de San Nicolas de Hidalgo (UMSNH) under the registration EBUM23538. The ethanolic extract was prepared following the methods described in *López-Mejía et al. (2019)*. The total yield was 22 g/100 g of fresh leaves, giving a percentage of 22% of crude extract of *C. citrinus*. The standardization of *C. citrinus* extract uses leaves of 4-year-old plants, age with the highest terpene content (*Petronilho et al., 2013*).

### Chemical materials

1,1-diphenyl-2-picrylhydrazyl, 2,2′-azinobis-3-ethylbenzothiazoline-6-sulfonic acid, 2,4,6-tri (2-pyridyl-s-triazine), 6-hydroxy-2,5,7,8-tetramethylchroman-2-carboxylic acid, Folin-Ciocalteu reagent, Gallic acid, aluminum chloride, Triton X-100, sodium deoxycolate, phenylmethylsulfonyl fluoride, protease inhibitor cocktail, 3,3′,5,5′-tetramethylbenzidine, arachidonic acid, Tumor necrosis factor alpha (TNF-α), interleukin-6 (IL-6), leptin, and adiponectin, purchased from Sigma-Aldrich (Merck/Mexico/Life Science Products & Service Solutions).

## DPPH radical assay

Free radical scavenging capacity was analyzed using the method of *Karamac, Kosiňska & Pegg (2005)*. The reduction of the 1,1-diphenyl-2-picrylhydrazyl (DPPH) radical was measured by monitoring the decrease of absorption at 517 nm. Briefly, the solution mixture contained: 100 µL of sample, 2,000 µL of methanol, 250 µL 1 mmol/L of DPPH solution. The mixture shakes for 15 s and incubated at room temperature in the dark for 20 min. After that, the absorbance was measured at 517 nm. The radical-scavenging activity (RSA) was calculated as a percentage of DPPH discoloration using the equation: % RSA = $100 \times (1 - A_C/A_D)$, where $A_C$ is the absorbance of the solution when the extract has been added at a particular level, and $A_D$ is the absorbance of the DPPH solution. 6-hydroxy-2,5,7,8-tetramethylchroman-2-carboxylic acid (Trolox) at (25–800 µm) was used as standard.

## ABTS radical scavenging assay

2,2′-azinobis-(3-ethylbenzothiazoline-6-sulfonic acid radical cation ($ABTS^{.+}$) was prepared as reported in *López-Mejía et al. (2021)*. 50 µL of sample was added to 950 µL of $ABTS^{.+}$ fresh solution with initial absorbance of 0.70 ± 0.02 at 734 nm, mixed and read during 7 min. Trolox range from 25 to 600 µm and TEAC was expressed as Trolox equivalents (TE)/g fresh weight (f.w).

## Ferric reducing antioxidant power (FRAP) assay

This assay is based in the reduction capacity of the antioxidant in reduction of ferric ion to produce a dark blue color ferrous complex. Working solution was prepared freshly and heated at 37 °C for 10 min. It contained: 10 mm 2,4,6-tri (2-pyridyl-s-triazine) (TPTZ) in 40 mm HCL, 20 mm ferric chloride ($FeCl_3.6\ H_2O$), and 300 mm sodium acetate buffer (pH 3.6) in a 1:1:10 ratio. 100 µL of sample was added 1,500 µL to working solution. The mixture was shaken and left to stand at room temperature for 20 min in the dark. Read absorption at 593 nm. Trolox was used as a standard ranged from 25 to 800 µm (*Thaipong et al., 2006*).

## Determination of total phenolic content

Phenolic content was measured by the formation of blue molybdenum complex using the Folin-Ciocalteu reagent. 200 µL of sample with 1,000 µL of Folin-Ciocalteu reagent (1:9 v/v) were vortexed for 5 min. Then, 1,000 µL of 7% $Na_2CO_3$ solution and 5,000 µL of distilled water were added. It was incubated at room temperature in the dark for 60 min, before measuring the absorbance at 765 nm. Gallic acid was used as standard (0.01–0.4 mm). Total phenolic content was expressed as mg gallic acid equivalent (mg GAE) (*Pripdeevech et al., 2010*).

## Total flavonoid content

Total flavonoids content was determined using the aluminum chloride colorimetric method reported by *Chang et al. (2002)*. A total of 500 µL of the sample were mixed with 1,500 µL of 95% methanol, 100 µL of 10% aluminum chloride, 100 µL 1 M potassium

acetate and 2,800 µL of distilled water. Then, stood for 30 min at room temperature in the dark, before measuring the absorbance at 415 nm. Water was used instead of aluminum chloride as blank. Rutin acid was used to calculate the standard curve (0.025–0.5 mg/mL).

## Total terpenoid content

The mixture contained: 100 µl of sample (10 mg/mL), 150 µl of vanillin/glacial acetic acid (5% w/v), and 20 µl of sulfuric acid. It was incubated at 60 °C for 45 min. The mixture was left on ice for 7 min to stop the reaction. Finally, 2,250 µl of glacial acetic acid was added, and its absorbance was measured at 548 nm. We used 1,8-Cineole at (1–6 mg/mL) as the standard (*Chang & Lin, 2012*).

## GC-MS analysis

The samples were analyzed in an Agilent 7890A gas chromatography equipment (Agilent Technologies, Santa Clara, CA, USA) with an HP5MS60M column (60 × 0.25 × 0.25; Agilent Technologies) coupled to an electronic impact ionization quadrupole mass analyzer mass spectrometer (Hewlett Packard 5975C). Operational condition was the same as reported in *Petronilho et al. (2013)*. Total ion chromatograms (TIC) were processed using the automated data processing software MSChem (Agilent Technologies).
To identify the different compounds, the mass spectrum of each compound detected was compared to those in mass spectral databases (Wiley 275 and US National Institute of Science and Technology (NIST) V. 2.0). The quantities of compounds were calculated from a standard calibration curve using 1,8-cineole at range (1–0.2 mg/mL).

## Animals

Nulliparous female Wistar rats (*Rattus norvegicus*), 4-months-old, 210–230 g, were collected from the animal production unit of the Animal House at the Chemical-Biological Research Institute, UMSNH. The animals were kept in the same house with the following standard conditions: 12 h light/12 h dark, relative humidity 60–70%, average temperature of 20 ± 3 °C, with access to food and water *ad libitum*. The care of the animals and the experimental process were approved by the Institutional Committee for Use of Animals of the UMSNH (approval date: 01/12/2021; protocol ID IIQB-CIBE-06-2021) and conducted according to the guide for the care and use of laboratory animals, Mexican Official Standard (NOM-062-ZOO-1999).

## Induction of obesity and gastric ulcers

### First stage

The animals had free access to water and food. Eighteen nulliparous female Wistar rats were randomly divided into three groups (*n* = 6 rats in each group). Group 1 (Control) received standard diet consisted of commercial chow (Rodent diet®). Group 2 (HFSD) (high fat and sugar diet) received 40% of standard food, enriched with 40% fat lard, 40% margarine (Aurrera®), and 20% sucrose (Zulka®) per 100 g food. Group 3 (HFSD + *C. citrinus*), fed with HFSD diet and *C. citrinus* leaf extract (250 mg/kg BW) administered daily by oral gavage. Previous studies (*López-Mejía et al., 2019*) showed that a daily single dose of *C. citrinus* extract leaves at 250 mg/kg was effective to reduce the number of tumors

in a 1,2-dimethylhydrazine-induced colon carcinogenesis in Wistar rats. Moreover, *Ortega-Pérez et al. (2022)* also found that a single dose reduced the body weight of Wistar rats fed with a high fat fructose diet. This diet has a caloric intake of 5.37 kcal/g (*Rahman et al., 2017*). The diet was prepared every week and kept at 6 °C for preservation. All animals were fed for 15 weeks. Food and water consumption were measured daily, and the animals' weight was recorded weekly to obtain the final BW gain at the end of the first stage of the experimental model.

### Second stage
At the end of 15 weeks of feeding the animals of groups two and three with HFSD, three additional control groups (*n* = 6), fed with standard diet, were included as follows:

Group 4, IND.

Group 5 (*C. citrinus* + IND), administered with a single dose of *C. citrinus* leaf extract (250 mg/kg).

Group 6 (Omeprazole (OME) + IND), administered with a single dose of OME (30 mg/kg).

At this stage all groups (1–6) were fasting for 12 h with access to water *ad libitum*. Then, 30 mg/kg of a single dose of IND was administered by oral gavage in groups 2, 3, 4, 5, and 6. After 4 h of administration of IND to groups 2, 3, 4, 5, and 6, the animals were slaughtered. The final weight of groups 1, 2, and 3 was determined at the end of the experimental model and all the animals were sacrificed by using a dose of sodium pentobarbital anesthesia (150 mg/kg) injection in peritoneal region. The blood was collected from abdominal aorta and placed into tubes at 4 °C for 30 min, then the clot was removed and samples were centrifuged at 3,000 rpm for 10 min at 20 °C to obtain the serum. Then, the stomach, liver, kidney, heart, and visceral fat of the rodents were removed, washed with 0.9% saline solution, weighed in digital scale, and stored at −20 °C.

## Determination of morphometric and biochemical parameters
The final weight of groups 1, 2, and 3 was determined at the end of the experimental model. The abdominal circumference, nose-anus (NAL) length, and nose-tail length were measured using a measuring tape. A digital weighing scale was used to obtain the total BW. The Lee index (g/cm) (equivalent to BMI) was calculated as LI = $(3\sqrt{BW})/NAL) \times 10$ (*Aslani et al., 2016*) and the percentage of body fat (%) was determined by the equation: 0.73 (LI-280.8). In this way, the percentage of fat only depends on the weight and nasal-anus length of the rats. The adiposity index (AI) was calculated as AI = (total adipose tissue weight/final BW) × 100. Visceral adipose tissue (VAT), which surrounds the abdominal organs, is the sum of subcutaneous, gonadal, mesenteric, retroperitoneal, and perirenal fat. Biochemical parameters in serum such as glucose, cholesterol, and triglycerides were determined using commercial SPINREACT® kits and following the specification established.

## Cytokine assays in stomach tissue
We homogenized a 100 mg piece of stomach tissue with 770 μL extraction buffer that contained 100 mm Tris-HCL pH 7.4, 150 mm NaCl, 1 mm EGTA, 1 mm EDTA,

1% Triton X-100, 0.5% sodium deoxycolate, 200 μL protease inhibitor cocktail, 10 μL phosphatase inhibitor cocktail 2, 10 μL phosphatase inhibitor cocktail 3, and 10 μL of serine protease inhibitor, such as 1 mm phenylmethylsulfonyl fluoride (PMSF). The sample was then placed in an orbital shaker with constant agitation for 1 h and 30 min at 4 °C. It was centrifuged for 12,000 rpm at 4 °C, and finally was placed on ice, and aliquots were made from the supernatant and stored at −80 °C. Tumor necrosis factor alpha (TNF-α), interleukin-6 (IL-6), leptin, and adiponectin were determined in the supernatant, using the enzyme-linked immunosorbent assay (ELISA) with Sigma-Aldrich kits®, following the manufacturer's specifications.

### Advanced oxidation protein products (AOPP) assay

AOPP were determined using the method by *Witko-Sarsat et al. (1996)*. The reaction mixture contained 1,000 μL of 20 mm phosphate buffer pH 7.4, 50 μL of stomach homogenate, and 50 μL of 1.16 M potassium iodide. We then added 100 μL of acetic acid. The mixture was centrifuged at $5,800 \times g$ for 5 min, and immediately after, the final absorbance was read at 340 nm. The AOPP content was determined using chloramine-T as the standard in calibration curve, at concentrations 5–100 μmol/L. AOPP concentration was expressed as μmol/L of chloramine-T equivalents.

### Myeloperoxidase (MPO)

MPO activity was performed using the protocols described by *Marquez & Dunford (1997)* with a slight modification. Briefly, stomach tissue was homogenized in 50 mm phosphate buffer pH 7.4 containing 0.5% of hexadecyltrimethyl ammonium bromide and then sonicated for 15 s. After that, the sample was frozen-thawed three times and centrifuged at 17,000 rpm for 20 min at 4 °C. The reaction mixture contained 425 μL of 200 mm phosphate buffer pH 5.4, 10 μL of 15 mm $H_2O_2$, and 40 μL of 20 mm of reagent 3,3′, 5,5′-tetramethylbenzidine (TMB). After mixing, 25 μL of stomach homogenate was added, incubated at 37 °C for 3 min in darkness, and placed on ice for 3 min. Finally, 1,000 μL of 200 mm sodium acetate pH 3 was added to stop the reaction, and the absorbance was measured at 655 nm for 3 min.

### Cyclooxygenase (COX-1) and COX-2 activity

Peroxidase activity of COX was determined *via* the transformation of acid arachidonic to prostaglandin G2 ($PGG_2$), followed by the quantitative conversion of $PGG_2$ to prostaglandin H2 ($PGH_2$) using the colorimetric substrate N,N′,N′,N′-tetramethyl-p-phenylenediamine (TMPD), as described in *Kumar et al. (2011)*. The reaction mixture contained 712 μL of 100 mm Tris-HCl buffer pH 8, 31 μL of 15 μm hematin, 31 μL od 3 μm EDTA, 100 μL of the previously centrifuged stomach homogenate (at $10,000 \times g$ for 15 min), and 63 μL of 100 mm TMPD. Finally, 63 μL of 133 μm arachidonic acid was added as a substrate, mixed, and incubated for 20 min at 25 °C, and the absorbance was measured at 590 nm. Simultaneously, one tube for each sample with inhibitor substrate (etoricoxib a selective COX-2 inhibitor) was prepared to determine COX-1 activity

differentially. TMPD extinction coefficient was 0.00826 $\mu M^{-1}$. One unit of enzyme was required to oxidize 1 nmol of TMPD per min.

### 5-Lipoxygenase (5-LOX)

The activity of 5-LOX was measured using a colorimetric inhibitor assay described in *Kumar et al. (2011)*. The assay is based on the generation of the complex $Fe^{3+}$/xylenol orange salt. The reaction mixture contained 490 µL of 50 mm Tris-HCl buffer pH 7.4, 10 µL of the previously centrifuged stomach homogenate (at 10,000 × g per 5 min), and 10 µL of 133 µm arachidonic acid. It was mixed and incubated at room temperature in darkness for 10 min. After that, 490 µL of FOX reagent was added. FOX reagent contained 25 mm sulfuric acid, 100 µm orange xylenol, and 250 µm ferrous sulphate, diluted in water-methanol (1:9). Finally, 100 µL of butylhydroxytoluene as antioxidant was included, mixed, and incubated as previously mentioned. Final absorbance at 590 nm was measured.

### Malondialdehyde (MDA) and Hydroxyalkenals (HNE)

To determine MDA, a reaction mixture of 200 µl homogenate, 5 µl of 5 mm butylated hydroxytoluene (BHT), 650 µl of 10 mm 1-methyl-2-phenylindole and 150 µl of 37% HCl was incubated at 45 °C for 60 min. Then it was kept on ice to stop the reaction, and measured at 586 nm. The total of HNE plus MDA was determined as mentioned before, replacing the hydrochloric acid with 37% methanesulfonic acid. The blank carries all components except homogenate sample. The levels of lipid peroxidation products were expressed in nmol MDA/g tissue and nmol HNE/g tissue (*Johnston et al., 2007*).

### Determination of gastric lesions by histological analysis

The stomach was opened along the greater curvature. Stomach contents were extracted, washed with 0.9% saline solution, and then fixed to a surface to extend them properly. The stomach damage was analyzed using ImageJ software (Wayne Rasband MD, USA) (*Tamaddonfard et al., 2019*; *Nabil et al., 2021*). The ulcer index (UI) in mm was assessed by measuring the lesion length and multiplying it by a severity factor (0 = no lesion, 1 =< 1, 2 = 2–4 and 3 => 4) (*Peskar, Ehrlich & Peskar, 2002*). The percentage of ulcer inhibition was determined as ((IU IND—IU group *C. citrinus*)/IU IND) *100) (*Shahin, Abdelkader & Safar, 2018*). A portion of the stomach was kept in 10% formaldehyde for fixation, and then the samples were rinsed with running water and dehydration was carried out every 15 min with three changes of 95% alcohol, three changes of absolute alcohol, one change of absolute alcohol/xylene, two changes in xylene, and three changes in liquid paraffin. Subsequently, paraffin inclusion was carried out and histological sections were made (5 µm thick) in a microtome (Leica®). Afterward, the samples were stained with hematoxylin and eosin (H&E).

Lesions induced by IND were observed with an optical microscope (Leica®) coupled to a camera at 100x magnification. Histological changes were evaluated as reported by *Luo et al. (2018)*. Briefly, 25 optic fields by section were observed to assign score values (higher values indicated more intense histopathological alteration) for: (a) epithelial cell loss (score 0–3), (b) hemorrhage (score 0–4), (c) lamina propria mucosae erosions

(score 0–4), (d) edema and disruption in the submucosa (score 0–4), and (e) inflammatory cell infiltration (score 0–3). Due to edema and immune cell recruitment constitute chief indicators of gastric ulcers, the score assignment was done according to previously described criteria in *Siriviriyakul et al. (2020)*. Briefly, a score of zero was assigned to histological fields with no infiltration of polymorphonuclear and mononuclear cells, a score of one to mild infiltration (1-5 polymorphonuclear and mononuclear cells), a score of two to moderate infiltration (6–10 polymorphonuclear and mononuclear cells), and a score of three to severe infiltration (more than 10 polymorphonuclear and mononuclear cells).

## Statistical analysis

The statistical analysis was generated in the JMP Pro version 14.0 program performing a one-way variance analysis (ANOVA) followed by *post hoc* Tukey test for morphological and biochemical parameters, enzymes, and biomarkers involved during inflammation. A *post hoc* LSD test was used to determine the ulcer index, inhibition %, and histopathological score with a standard mean error (statistically significant at $p \leq 0.05$). The graphs were elaborated in the GraphPad Prism version 8.0 program. Before using one-way ANOVA test, the assumption of normal distribution of the data was evaluated by applying the Kolmogorov-Smirnov test and the homogeneity of variance was determined using the Barlett test. In both cases, a significance level of 5% was used.

# RESULTS

## Chemical analysis of ethanolic leaf extract of *C. citrinus*

Antioxidant capacity and the quantification of total phenols, flavonoids, and terpenes were similar to the values reported in *Ortega-Pérez et al. (2022)*. The characterization of ethanolic leaf extract of *C. citrinus* was also performed to analyze and identify terpenes compounds using GC/MS (Table 1).

## Induction of obesity (morphological and biochemical parameters)

Contrary to the control and HFSD + *C. citrinus* groups, the HFSD group fed with the HFSD had a significant increase ($p < 0.05$) in weight gain after 15 weeks. Figure 1 shows that the final BW in the HFSD group was 22% higher than that of the control group. Conversely, the final BW in the HFSD + C. *citrinus* group increased only 7%, as compared to the control group.

Although the HFSD group showed a reduction in food and water intake as compared to the control group, it presented an increase of 5.6 times in fat and 2.6 times in visceral fat deposition (Table 2). The HFSD group had a Lee index of 0.310 (any index greater than 0.3 is considered obese). On the contrary, the HFSD group treated with *C. citrinus* only increased 3.4 times in fat and 2.0 times in visceral fat deposition after 15 weeks (Table 2). In addition, the stomach, liver, kidney, and heart weights were greater in the HFSD group than in the control and HFSD + *C. citrinus* groups. A statistically significant increase was observed in the levels of glucose, cholesterol, and triglycerides in the HFSD group when compared with the control and HFSD + *C. citrinus* groups. The oral administration of

**Table 1 Parameters evaluated to standardize ethanolic extract of 4-year-old leaves of *Callistemon citrinus*.**

| $EC_{50}$ values (µg/ml) | |
| --- | --- |
| DPPH scavenging activity | 84.00 ± 0.20 |
| FRAP scavenging activity | 11.76 ± 0.11 |
| Trolox scavenging activity | 150.00 ± 0.25 |
| Total phenols, flavonoids and terpenes contents | |
| Phenols (GAE mg/g fw) | 247.10 ± 3.07 |
| Flavonoids (Rutin (mg/g fw) | 133.87 ± 1.09 |
| Terpenes (α-Terpineol mg/g fw) | 90.60 ± 0.22 |
| Major terpenes identified using GC-MS | |
| Compounds | (µg/ml) |
| α-Pinene | 102.06 ± 22.03 |
| α- Phellandrene | 38.33 ± 4.89 |
| Limonene | 321.15 ± 34.14 |
| α-Terpinolene | 38.63 ± 8.90 |
| 1,8-Cineole | 197.57 ± 10.08 |
| Linalool | 32.39 ± 7.67 |
| 4-Terpineol | 58.95 ± 11.06 |
| α-Terpineol | 68.89 ± 8.54 |
| Aromadendrene | 61.29 ± 5.32 |
| Spathulenol | 14.03 ± 0.53 |

**Note:**
Values given as mean ± SD ($n = 6$).

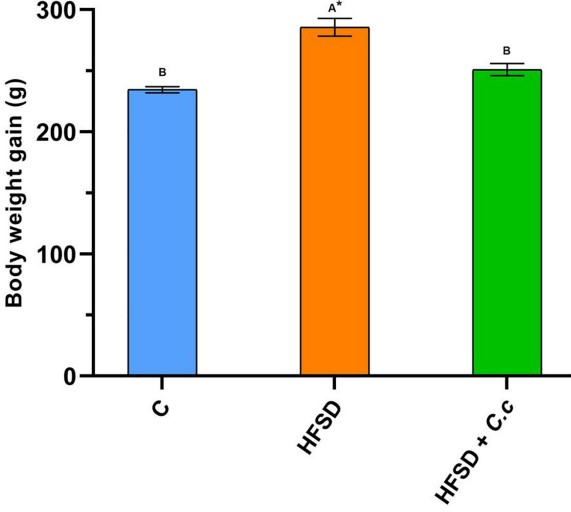

**Figure 1 Final weight gain of the groups fed for 15 weeks with a diet high in fat and sugar (HFSD) and extract of *C. citrinus* (250 mg/kg) (*C.c*), and control group (C).** Data shown as mean ± S.E.M. (ANOVA one-way followed by Tukey test, values statistically different ([A,B]) among groups (*$p < 0.05$), $n = 6$).

**Table 2 Effect of *Callistemon citrinus* on morphological and biochemical parameters in Wistar rats fed with a high fat-sugar diet.**

| Parameters | Control | HFSD | HFSD + *C. citrinus* |
|---|---|---|---|
| Initial weight (g) | 215.4 ± 4.60[A] | 228.7 ± 6.98[A] | 211.8 ± 9.60[A] |
| Final weight gain (g) | 234 ± 30[B] | 286 ± 70[A]* | 251 ± 50[B] |
| Abdominal circumference (cm) | 20 ± 0.20[B] | 21 ± 0.50[A]* | 20 ± 0.20[B] |
| Length nose-anus (cm) | 21 ± 0.10[B] | 22 ± 0.10[A]* | 21 ± 0.30[B] |
| Percentage of fat (%) | 5 ± 0.90[A] | 28 ± 3.80[B]* | 17 ± 1.00[C]* |
| Adiposity index (%) | 3.6 ± 0.59[C] | 12.0 ± 0.5[A]* | 10 ± 0.51[B]* |
| Lee index (g/cm) | 0.29 ± 0.02[B] | 0.31 ± 0.02[A]* | 0.30 ± 0.03[A, B] |
| Visceral fat deposition (g) | 13 ± 0.60[A] | 34 ± 1.90[B]* | 26 ± 0.60[C]* |
| Stomach (g) | 1.6 ± 0.04[B] | 1.8 ± 0.02[A]* | 1.5 ± 0.03[B] |
| Liver (g) | 7.1 ± 0.50[B] | 9.2 ± 0.10[A]* | 7.7 ± 0.10[B] |
| Kidney (g) | 1.9 ± 0.02[B] | 2.3 ± 0.05[A]* | 1.9 ± 0.01[B] |
| Heart (g) | 1.16 ± 0.0[B] | 1.34 ± 0.07[A]* | 1.02 ± 0.03[B] |
| Food intake (g/day) | 16 ± 0.30[A] | 13 ± 0.30[B]* | 10 ± 0.20[C]* |
| Water intake (ml/day) | 37 ± 0.60[A] | 31 ± 1.80[B]* | 23 ± 0.80[C]* |
| Glucose (mg/dL) | 120 ± 5.30[B] | 151 ± 9.53[A]* | 108 ± 3.82[B] |
| Cholesterol (mg/dL) | 51 ± 2.30[B] | 75 ± 2.40[A]* | 58 ± 4.51[B] |
| Triglycerides (mg/dL) | 134 ± 4.61[B] | 162 ± 1.22[A]* | 144 ± 2.04[B] |

**Note:**
Data shown as mean ± S.E.M. (ANOVA one-way followed by Tukey test, values statistically different ([A,B,C]) among groups (*$p < 0.05$), $n = 6$.

*C. citrinus* reduced all morphological and biochemical parameters in comparison with the HFSD group.

## Development of gastric lesions (macroscopes and histological analyses)

Administration of IND at a single oral dose of 30 mg/kg produced severe ulcer injuries and visible hemorrhagic lesions in the gastric mucosa, with an ulcer index of 14 ± 0.41 mm. Moreover, the HFSD + IND group presented with mucosal damage and an ulcer index of 8 ± 0.29 mm. On the other hand, administration of *C. citrinus* for 15 weeks and oral pretreatment with *C. citrinus* and OME at a single dose of 250 and 30 mg/kg, respectively, significantly decreased gastric lesions (Table 3).

Figure 2 shows that the control group (Fig. 2A) did not present with macroscopic lesions, whereas the HFSD + IND (Fig 2B) and indomethacin (Fig. 2C) groups displayed severe damage to the gastric mucosa with ulcer injuries and hemorrhagic lesions. In contrast, the ulcer lesion was improved significantly in the other groups. These results showed that *C. citrinus* and OME accelerated the healing of gastric ulcers in rats, who presenting with a decrease in the lesion area.

Results of the histological evaluation (Fig. 3) indicated a normal architecture of the gastric mucosal wall with no signs of damage in the control group (Fig. 3C). Conversely, the HFSD + IND and IND groups showed many areas with necrosis, cell death, and the infiltration of immunological cells. On the other hand, the HFSD + *C. citrinus* + IND

**Table 3 Rate of ulceration in the gastric mucosa and percentage of inhibition of lesions in the stomach of rats fed with a high-fat-sugar and administered with indomethacin.**

|  | Ulcer index (mm) | Percentage of inhibition of ulcers (%) | $p$ (*) |
|---|---|---|---|
| Control | – | – | D × I |
| HFSD + Ind | 8 ± 0.40[A*] | 41 ± 0.95[A*] | D × I |
| HFSD + *C. citrinus* + Ind | 3 ± 0.34[B] | 71 ± 1.10[B] | D × I |
| Indomethacin | 14 ± 0.37[A*] | – | I |
| *C. citrinus* + Ind | 5 ± 0.37[B] | 67 ± 1.07[B] | I |
| Omeprazole + Ind | 5 ± 0.37[B] | 70 ± 1.07[B] | I |

**Note:**
Data shown as mean ± S.E.M. (ANOVA one-way followed by LSD test, values statistically different ([A,B]) among groups (*$p < 0.05$), $n = 6$). D, diet; I, indomethacin; D × I, Diet × indomethacin.

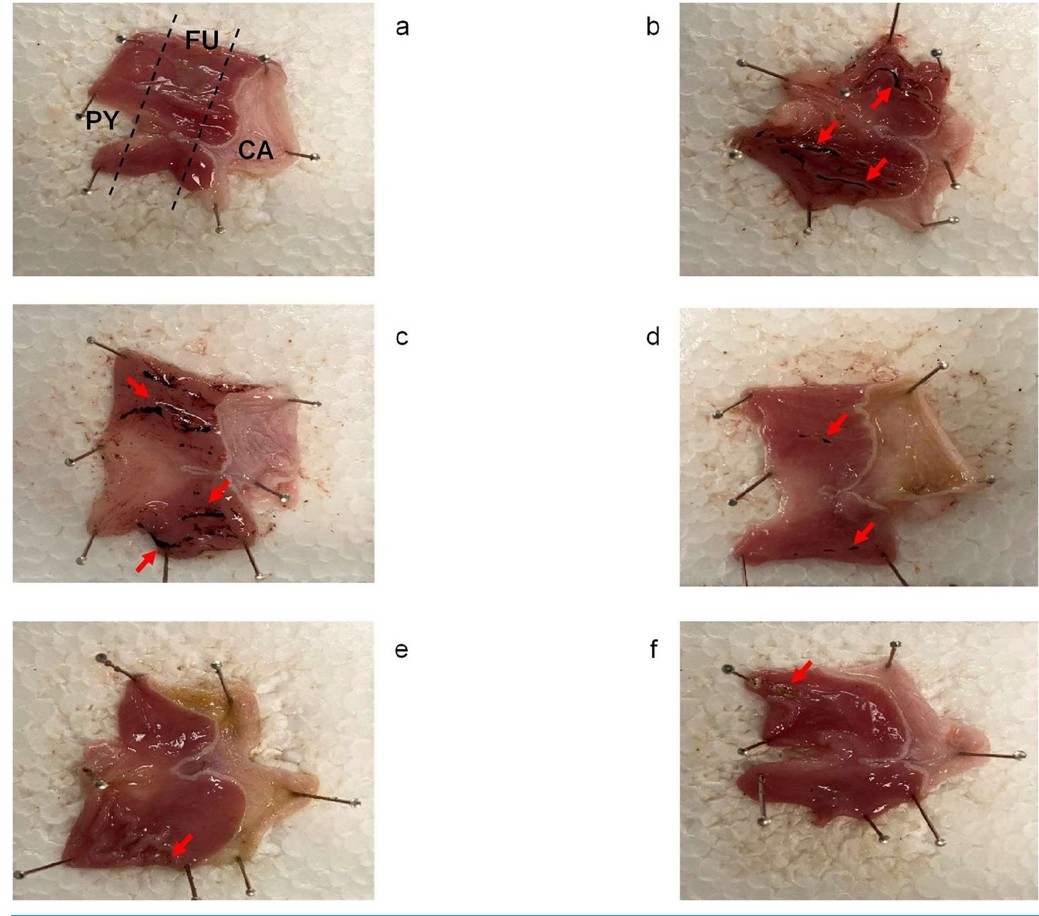

**Figure 2 Gastroprotective effect of *Callistemon citrinus*.** Rat stomach showing protective effect of *Callistemon citrinus* (*C.c*) on the indomethacin (IND) induced gastric ulcer in Wistar rat. Control (A); HFSD + IND (B); IND (C); HFSD + *C.c* 250 mg/kg + IND (D); *C.c* 250 mg/kg + IND (E); Omeprazole (OME) + IND (F). Indomethacin was administrated at 30 mg/kg for a single dose in B to F. CA, Cardial region; FU, Fundus region and PY, Pyloric region. (A) showed intact gastric mucosa. (B and C) presented several dark red submucosal regions (red arrows). In (D–F) showed a normal gastric mucosa with few lesions.

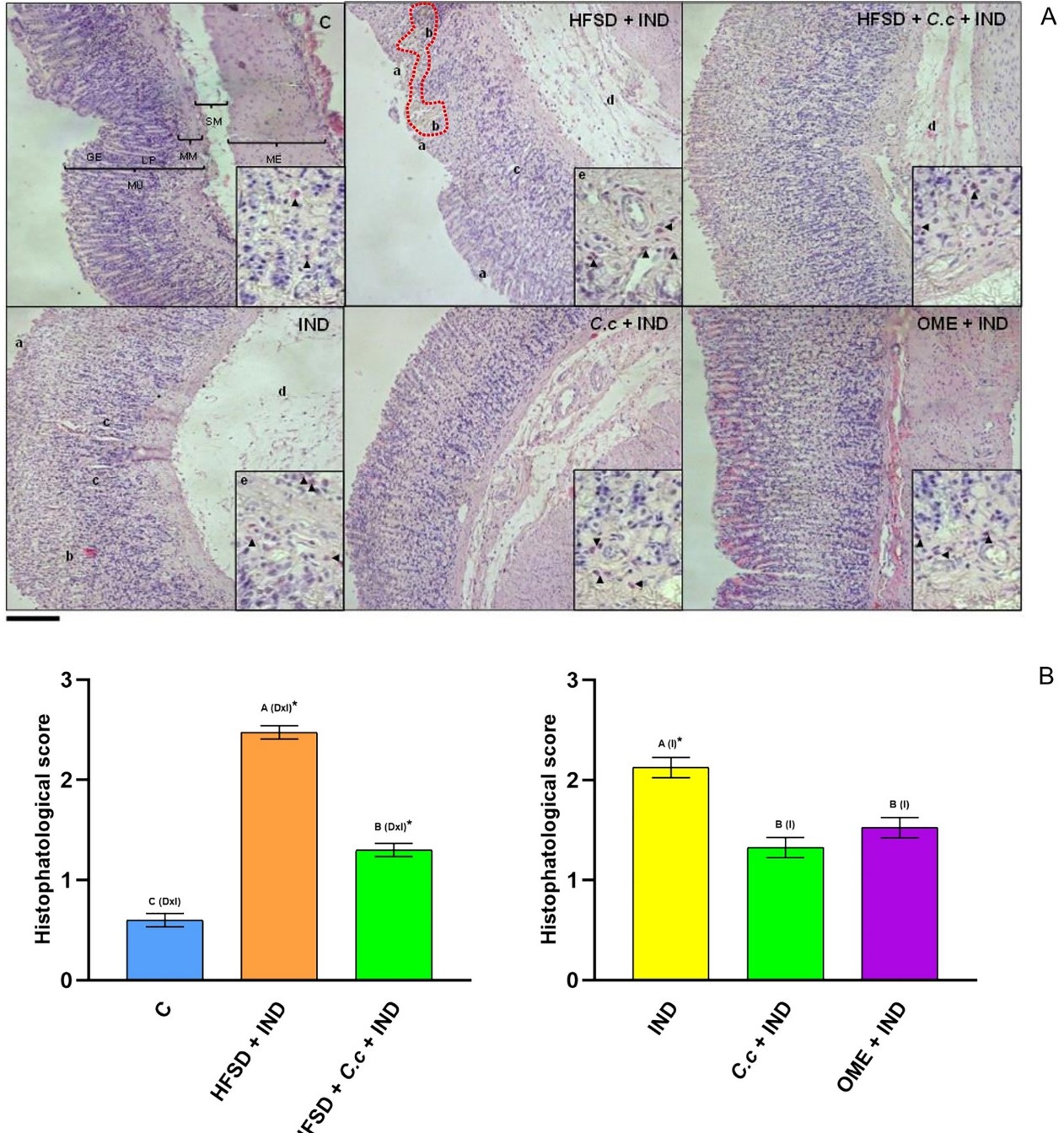

**Figure 3 Effect of *Callistemon citrinus* on gastric lesions generated by indomethacin (IND) in rats fed with a high-fat diet (HFSD). Microphotographs show representative cuts of the fundic region.** (A) The graph shows the values of the histopathological score that considers jointly the epithelial cell loss of the gastric mucosa (a), hemorrhage (b), the inflammatory cell infiltration (c), the lamina propria mucosae erosions (d), and the edema and disruption in the submucosa (e). Gastric mucosa (MU), Gastric epithelium (GE), Muscularis mucosae (MM), Submucosal (SM), external muscular (ME) and lamina propria (LP). Scale bar = 200 μm. (B, left) The control rats fed with normal diet, one group fed with HFSD and other group fed with HFSD + *C. citrinus*, chronically treated during 15 weeks; after this period, one single dose of indomethacin was given for the last two groups. (B, right) One group treated with a single dose of indomethacin, one group treated with a single dose of indomethacin + *C. citrinus* and the last group treated with a single dose of indomethacin + omeprazole. Data shown as mean ± S.E.M. (ANOVA one-way followed by Tukey test, values statistically different ([A,B,C]) among groups (*$p < 0.05$*), $n = 6$). Score assignment was done according to previously described criteria (*Siriviriyakul et al., 2020*).

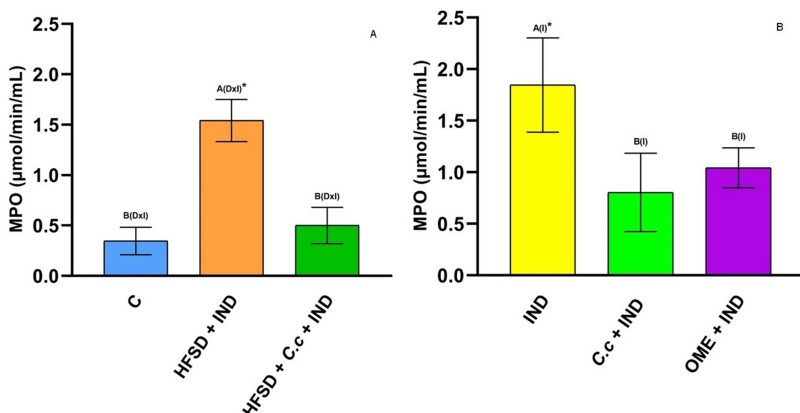

**Figure 4 Myeloperoxidase activiy.** Enzyme activity of myeloperoxidase in experimental groups. (A) The control rats fed with normal diet, one group fed with HFSD and other group fed with HFSD + *C. citrinus*, chronically treated during 15 weeks; after this period, one single dose of indomethacin was given for the last two groups. (B) One group treated with a single dose of indomethacin, one group treated with a single dose of indomethacin + *C. citrinus* and the last group treated with a single dose of indomethacin + omeprazole. Data shown as mean ± S.E.M. (ANOVA one-way followed by Tukey test, values statistically different ($^{A,B}$) among groups ($^*p < 0.05$), $n = 6$).

group presented with attenuated damage in the gastric mucosa, caused by IND, showing the beneficial effect of the extract. It also maintained the structural composition of the stomach layers, reduced gastric tissue necrosis and immune cell infiltration as well. The groups administered with a single dose of *C. citrinus* extract + IND and OME (OME + IND) had the ability to reduce gastric alterations caused by IND, displaying mucosal integrity of a treated animal similar to the control group.

## Effect of *C. citrinus* extract on enzymes involved during inflammation

Figure 4 shows a significant increase of MPO activity ($p < 0.05$) in the gastric mucosa of the HFSD + IND and IND groups, in contrast to the *C. citrinus* + IND and OME + IND groups. On the other hand, the group fed with HFSD + *C. citrinus* extract for 15 weeks, followed by administration of IND, the activity of the MPO decreased significantly ($p < 0.05$) as compared to the HFSD + IND and the IND groups. MPO activity is used as indicator of infiltration and neutrophil accumulation in gastric mucosa. A decrease of MPO activity in the group with *C. citrinus* treatment for 15 weeks, agrees with the reduced of neutrophil influx showed in Fig. 3.

Cyclooxygenase (COX) is an enzyme involved in the synthesis of prostaglandins (PGs). Fig. 5A shows a reduction of COX-1 in the HFSD + IND group, and the HFSD + *C. citrinus* + IND group maintained similar COX-1 activity to the control group. However, the COX-1 activity in groups administrated with *C. citrinus* + IND and OME + IND was similar to the IND group (Fig. 5B). On the contrary, the HFSD + IND group showed a higher increase in COX-2 activity, while the HFSD + *C. citrinus* + IND showed similar activity as the control group (Fig. 6A). On the other hand, the IND group had increased COX-2 activity unlike the *C. citrinus* + IND and OME + IND groups (Fig. 6B). Moreover, the

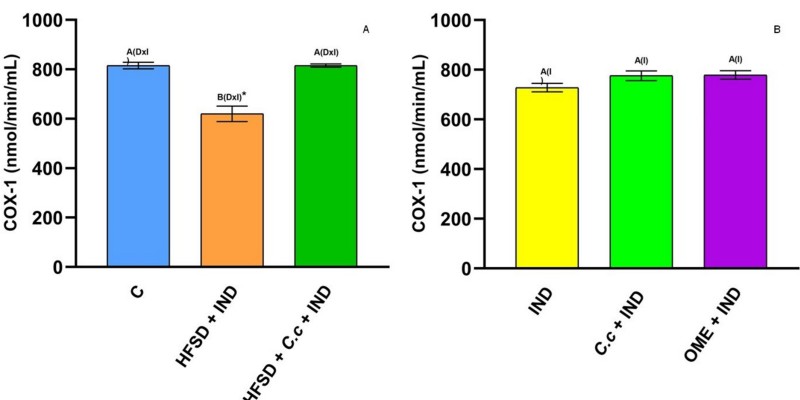

**Figure 5 Cyclooxygenase-1 activity.** Effect of indomethacin and *C. citrinus* extract on the activity of COX-1. (A) The control rats fed with normal diet, one group fed with HFSD and other group fed with HFSD + *C. citrinus*, chronically treated during 15 weeks; after this period, one single dose of indomethacin was given for the last two groups. (B) One group treated with a single dose of indomethacin, one group treated with a single dose of indomethacin + *C. citrinus* and the last group treated with a single dose of indomethacin + omeprazole. Data shown as mean ± S.E.M. (ANOVA one-way followed by Tukey test, values statistically different ($^{A,B}$) among groups ($^*p < 0.05$), $n = 6$).

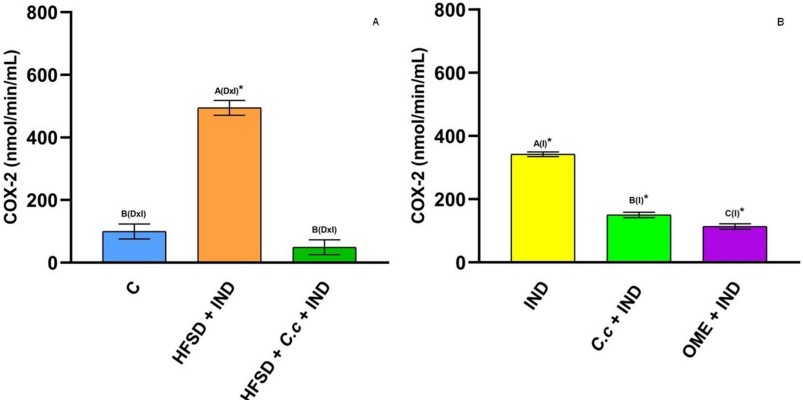

**Figure 6 Cyclooxygenase-2 activity.** Effect of indomethacin and *C. citrinus* extract on the activity of COX-2. (A) The control rats fed with normal diet, one group fed with HFSD and other group fed with HFSD + *C. citrinus*, chronically treated during 15 weeks; after this period, one single dose of indomethacin was given for the last two groups. (B) One group treated with a single dose of indomethacin, one group treated with a single dose of indomethacin + *C. citrinus* and the last group treated with a single dose of indomethacin + omeprazole. Data shown as mean ± S.E.M. (ANOVA one-way followed by Tukey test, values statistically different ($^{A,B,C}$) among groups ($^*p < 0.05$), $n = 6$).

HFSD + IND and IND groups presented the highest number of gastric ulcers (Fig.1), which supports the results of the COX activities.

5-LOX is an enzyme involved in inflammation processes through the synthesis of leukotrienes (pro-inflammatory mediators) and lipoxins (anti-inflammatory mediators). In this study, IND induction caused a significant increase ($p < 0.05$) of 5-LOX activity in the HFSD + IND and IND groups as compared to the other groups (Figs. 7A and 7B). Again, the 5-LOX activity is directly related to inflammation presented in gastric ulcers.

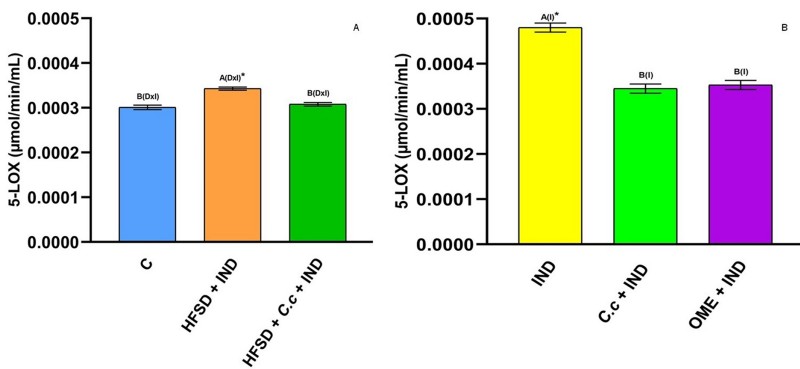

**Figure 7** **5-Lipooxygenase activity.** Effect of indomethacin and *C. citrinus* extract on the 5-LOX activity. (A) The control rats fed with normal diet, one group fed with HFSD and other group fed with HFSD + *C. citrinus*, chronically treated during 15 weeks; after this period, one single dose of indomethacin was given for the last two groups. (B) One group treated with a single dose of indomethacin, one group treated with a single dose of indomethacin + *C. citrinus* and the last group treated with a single dose of indomethacin + omeprazole. Data shown as mean ± S.E.M. (ANOVA one-way followed by Tukey test, values statistically different ($^{A,B}$) among groups ($^*p < 0.05$), n = 6).

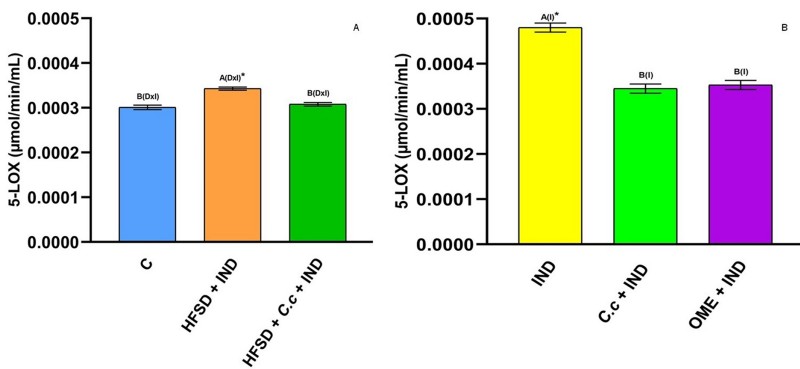

**Table 4 Biomarkers of stress oxidative and cytokines levels in the stomach of rats fed with a diet high in fat and sugar and administered with indomethacin.**

|  | Leptin (mg/dl) | Adiponectin (mg/dl) | AOPP (µmol/L) | IL-6 (mg/dl) | TNFα (mg/dl) | MDA (nM//mg protein) | HNE (nM/mg protein) | p(*) |
|---|---|---|---|---|---|---|---|---|
| **Control** | 304. 2 ± 12.8$^{C*}$ | 27.3 ± 0.9$^A$ | 3.8 ± 0.1$^B$ | 90.4 ± 5.9$^B$ | 2197.2 ± 28.2$^{C*}$ | 0.05 ± 0.009$^B$ | 0.05 ± 0.01$^B$ | D × I |
| **HFSD + Ind** | 904.7 ± 12.8$^{A*}$ | 13.9 ± 0.9$^{B*}$ | 6.4 ± 0.1$^{A*}$ | 179.5 ± 6.9$^{A*}$ | 3489.4 ± 28.2$^{A*}$ | 0.19 ± 0.009$^{A*}$ | 0.15 ± 0.01$^{A*}$ | D × I |
| **HFSD + *C. citrinus* + Ind** | 454.6 ± 12.8$^B$ | 31.0 ± 0.9$^A$ | 2.2 ± 0.1$^{C*}$ | 116.6 ± 6.9$^B$ | 3058.9 ± 28.2$^B$ | 0.08 ± 0.009$^B$ | 0.07 ± 0.01$^B$ | D × I |
| **Indomethacin (Ind)** | 430.1 ± 14.4$^{A*}$ | 17.6 ± 0.2$^{B*}$ | 4.9 ± 0.1$^{A*}$ | 121.7 ± 6.4$^{B*}$ | 2455.7 ± 76.5$^{B*}$ | 0.14 ± 0.005$^{A*}$ | 0.09 ± 0.01$^{A*}$ | I |
| ***C. citrinus* + Ind** | 348.8 ± 14.4$^B$ | 12.3 ± 0.2$^{C*}$ | 3.6 ± 0.1$^B$ | 97. 6 ± 6.4$^B$ | 2445.0 ± 76.5$^B$ | 0.12 ± 0.005$^{A,B}$ | 0.06 ± 0.01$^B$ | I |
| **Omeprazole + Ind** | 490.9 ± 14.4$^A$ | 19.4 ± 0.2$^A$ | 4.8 ± 0.1$^A$ | 154.2 ± 6.4$^A$ | 4410.5 ± 76.5$^A$ | 0.11 ± 0.005$^B$ | 0.03 ± 0.01$^{C*}$ | I |

**Note:**
Data shown as mean ± S.E.M. (ANOVA one-way followed by Tukey test, values statistically different ($^{A,B,C}$) among groups ($^*p < 0.05$), n = 6). D, diet; I, indomethacin; D × I, Diet × indomethacin.

However, the HFDS + *C. citrinus* + IND, *C citrinus* + IND, and OME + IND groups showed a significant reduction in 5-LOX activity.

### Effect of *C. citrinus* on biomarkers of oxidative stress and cytokines levels

Table 4 shows that the HFSD group increased the levels of leptin, advanced products of protein oxidation, interleukin 6, and tumor necrosis factor alpha, as compared with the control group, whereas a minor decrease ($p < 0.05$) on adiponectin was observed. Daily administration of *C. citrinus* leaf extract for 15 weeks followed by IND significantly

decreased ($p < 0.05$) leptin, TNFα, and IL-6 levels and significantly increased ($p < 0.05$) adiponectin levels as compared to the HFSD + IND group. IND produced high levels of MDA and HNE (lipid peroxidation products) and AOPP in all groups with the exception of the HFSD + *C. citrinus* + IND group. However, the HFSD + IND group presented with the highest HNE levels, suggesting that the diet also had damaging effects. Animals treated with IND, *C. citrinus* + IND, and OME + IND presented with similar biomarker values. Moreover, the HFSD group treated with *C. citrinus* for 15 weeks showed a significant decrease in all the biomarkers in comparation to the HFSD group.

## DISCUSSION

Obesity is associated with diabetes mellitus type 2, hypertension, cardiovascular morbidity and gastrointestinal diseases (*Emerenziani et al., 2020*). It is also considered a chronic low-grade systemic inflammation state (*Lin & Li, 2021*). Moreover, the intake and prolonged use of NSAIDs cause gastric ulcers and gastrointestinal disorders (*Wallace, 2019*). This is the first study that shows the gastroprotective effects of the *C. citrinus* leaf extract in the model of IND-induced gastric ulcers and the relationship between a HFSD and the predisposition to more damage in the gastric mucosa. This study highlights the mechanism of the gastroprotective effect of *C. citrinus*.

The drugs used to control gastric ulcer are proton pump inhibitor (OME), antacids (aluminum hydroxide and magnesium hydroxide), anticholinergics (pirenzepine), and histamine receptor antagonists (famotidine). However, many of them have adverse effects (*Kim et al., 2019*). Natural products have the advantage of presenting with few side effects. Several studies on plant extracts with antiulcer activity demonstrated anti-inflammatory activity and protective action on the gastrointestinal mucosa through the muscularis mucosae, inhibited gastric secretion, increased mucus secretion, promoted healing, improved oxidative status through the reduction of MPO and COX-2, and increased inhibition of free radicals and lipid peroxidation (*Oliveira et al., 2014*). *Shareef et al. (2022)* reported that some medicinal plants have the ability to decrease ulcer areas, prevent edema, and inhibit leucocyte infiltration of the submucosal layer. The extract's effects are not limited to its presence in the stomach, since the bioavailability of the extract compounds systemically circulates in the blood, which ensures that it reaches all gastric layers through splanchnic circulation and there are no impediments to reaching the deeper layers (muscle mucosa or muscularis mucosa).

There is currently controversy over whether female hormones are involved in gastric protection. *Périco et al. (2019)* reported that the hydroalcoholic extract of the leaves of *E. punicifolia* had gastric healing effects, and these effects could be affected by female sex hormone interference. Additionally, gastric mucosal blood flow has an important role in mucosal protect against injuries (*Shore et al., 2017*), the authors showed that male rats had higher levels of blood flow than female rats, suggesting that female hormones have an effect on the predisposition to produce ulcers in female animals. Moreover, *Akpamu et al. (2016)* reported that the administration of testosterone in female rats was a factor to avoid damage in gastric mucosal.

On the other hand, *Uslu et al. (2002)*, concluded that stress ulcer formation was not influenced by gender or the estrus cycle of female rats. In another study investigating the effects of progesterone on FSH-stimulated IND ulcers in rats, it was suggested that low doses (1 mg/kg) of progesterone (which inhibits endogenous FSH) cannot sufficiently stimulate its own receptors for ulcer formation. However, at low doses, it may prevent the ulcerogenic effects of FSH by decreasing FSH concentration, which increases after ovariectomy. Additionally, progesterone is not an antiulcer hormone; on the contrary, it produces ulcers *via* its own receptor, and FSH may produce ulcerogenic effects *via* progesterone receptors (*Borekci et al., 2008*). *Sangma, Jain & Mediratta (2014)* found that female sex steroids neither worsened nor protected against gastric lesions. Finally, *Fonseca da Silva et al. (2020)* found that normotensive and hypertensive male and female rats responded equally in the development of gastric ulcers induced with IND. The involvement of female hormones in the gastric ulcers has not yet been firmly established.

Furthermore, this study showed the effect of HFSD consumption + IND on the early appearance of gastric ulcers as compared with a normal diet + IND. The administration of *C. citrinus* leaf extract prevented the appearance of the gastric ulcers. A pretreatment with one dose of *C. citrinus* extract also reduced the ulcer index to 8 mm, instead of the 14 mm that was seen in the IND and HFSD + IND groups. Our results agree with those reported by *Chanudom & Tangpong (2015)*, who used *Syzygium cumini* (L.) Skeels (Myrtaceae) extract on male mice with gastric ulcers induced by IND.

*Rocha Caldas et al. (2015)* showed that 1,8-cineole (100 mg/kg) presented with a 2.5 ± 1.2 mm gastric lesion area in an ethanol-induced ulcer in Wistar rats. The major compound of *Citri reticulatae* essential oil is limonene, it produces a reduction of the ulcer index in a model of HCl/EtOH-induced gastric ulcers, revealing its gastroprotective and healing effects (*Li et al., 2023*). *Moraes et al. (2009)* showed that an essential oil from *Citrus aurantium* (250 mg/kg) and limonene (245 mg/kg) showed stronger protection in the gastric mucosa of male mice with gastric ulcers induced with ethanol or IND by increasing gastric mucus and reducing the $H^+$ secreted on the gastric lumen. *Souza et al. (2011)* reported that the oral administration of α-terpineol at a dose of 10 mg/kg reduced the ulcer index in Wistar male rats with gastric ulcers induced with ethanol and IND. Despite the fact that our study used female Wistar rats, the results were similar. *López-Mejía et al. (2021)* showed that 250 mg/g fresh weight (fw) of *C. citrinus* leaf extract had a concentration of 0.88 mg/g fw of 1,8-cineole, 4.38 mg/g fw of limonene, and 3.28 mg/g fw. The concentrations of terpenes were lower than those used in other studies, suggesting a synergism among the compounds found in *C. citrinus* that have a gastroprotective effect.

The histological evaluation showed that IND caused inflammation of the gastric mucosa manifested with a high cell infiltration in the HFSD + IND and IND groups. Figure 3 showed that *C. citrinus* suppressed gastric inflammatory cell infiltration. Moreover, IND increased MPO activity.

The activity of MPO is related to the inflammatory process in gastric ulcers. However, the prolonged administration of *C. citrinus* extract, as well as a single dose, promote the decrease of enzymatic MPO activity (Fig. 5). Our results are similar to those reported for

*Dracocephalum kotschyi* (*Minaiyan et al., 2021*) and *Byrsonima intermedia* (*de Cássia Dos Santos et al., 2019*), plants with anti-inflammatory and antioxidant effects that decrease MPO activity, reduce the synthesis of $PGE_2$, and reduce lipid peroxidative products (biomarkers of oxidative stress). *Moraes et al. (2009)* reported that limonene has anti-ulcerogenic activity *via* induced mucus production and maintenance of $PGE_2$ levels similar to normal rats. This result corresponds to those of *Rozza et al. (2011)* that showed a reduction of $PGE_2$ levels by using limonene in IND-induced gastric ulcer in rats. *de Souza et al. (2019)* reported the protective effect of limonene against ethanol-induced gastric ulcers in rats *via* the suppression of MPO expression. *Bose et al. (2021)* reported the effects of α-terpineol to reduce inflammatory marker levels as MPO in the keratitis process induced by *Pseudomonas aeruginosa*. Our results are consistent with those studies reporting that 1,8-cineole has gastroprotective and healing effects by increasing the production of mucus and reducing MPO activity and lipid peroxidation products (*Périco et al., 2020*).

NSAIDS induce gastric ulceration by blocking the activity of COX-1 and COX-2, which produces endogenous PGs and reduces secretion of mucus and bicarbonate. They also decrease mucus blood flow and impaired platelet aggregation (*Wallace, 2019*). A high fat diet can stimulate innate immune cells and set an inflammatory status (*Gil-Cardoso et al., 2017*). *C. citrinus* showed a higher percentage of COX-2 inhibition than IND; this NSAID is a nonselective COX inhibitor that inhibits COX-1 and gastric damage (*Fabri et al., 2013*). Our study showed a high activity of COX-1 in HFSD + *C. citrinus* + IND, *C. citrinus* + IND, and OME + IND groups, as compared to the control group. The relationship between COX-1 activity and gastric damage was also reported by *Antonisamy et al. (2016)*. *Azab, Jaleel & Eldahshan (2017)*, found that *Cinnamomum glanduliferum* oil ameliorates gastric damage in rats using an ethanol-induced model that decreases COX-2 activity. The major compounds were 1,8-cineole, sabinene, α-terpineol, and α-pinene. *C. citrinus* presented a high inhibition of COX-2, with the main compounds 1,8-cineole and α-terpineol. This inhibitory activity could be attributed to the presence of the monoterpenes. Conversely, limonene prevented an inflammatory process by reducing oxidative stress biomarkers and the expression of COX-2 in kidneys of rats treated with doxorubicin (*Rehman et al., 2014*).

COX and LOX are considered a rate-limiting enzyme in the inflammatory process. Inhibition of these enzymes prevents the synthesis of PGs and leukotrienes mediators involved in an inflammation process (*Mukhopadhyay et al., 2023*). 5-LOX is activated when COX-1 is blocked. Leukotrienes are mediators that activate leukocyte adherence and contribute to mucosal injury (*Martel-Pelletier et al., 2003*). Our results showed that *C. citrinus* reduced COX-2 and 5-LOX activities. This attenuation may control the vascular changes during inflammation (*Santos & Rao, 2001*), suggesting a potential anti-inflammatory property in *C. citrinus* leaves.

Production of pro-inflammatory cytokines impairs the inflammation of gastric mucosa (*Wei et al., 2021*). This is consistent with our results that showed high TNF-α, IL-6, and leptin levels in the HFSD + IND and IND groups, suggesting that an increase in these parameters is related to ulcer gastric injury and HFSD intake. Previously, *Ayala-Ruiz et al.*

(2022), showed that 1,8-cineole, limonene, α-terpineol, and terpenes mixture reduced oxidative stress in the liver of rats fed with a HFSD by decreasing TNF-α, IL-6, and leptin levels and also reducing biomarkers of oxidative stress.

Lipid peroxidation products (MDA and HNE) are considered biomarkers of oxidative stress in membranes, whereas MDA is a mutagenic product. HNE is more toxic because of its fast reaction with amino and thiols groups (*Ayala, Muñoz & Argüelles, 2014*). *Miura, Muraoka & Fujimoto (2002)*, reported that gastric mucosa damage caused by IND involved lipid peroxidation. Therefore, high MDA and HNE levels are present during a gastric injury. The relationship between the increase of lipid peroxidation products and IND has been reported by *Lim et al. (2019)*; in this study, the IND dose was 25 mg/kg. In our study the IND dose was 30 mg/kg. The increase in MDA level in the HFSD + IND and IND groups is related to high gastric damage (Fig. 3).

The activation of neutrophil by IND causes an increase of the reactive oxygen species (ROS) levels in the gastric mucosa. The ROS can damage biomolecules as protein, lipids, and DNA. The AOPP have been used as a marker of oxidative stress since the proteins are targets for oxidants. *Witko-Sarsat et al. (1996)* found a correlation between AOPP and protein damage. High levels of AOPP have been found in diabetes mellitus, cardiovascular disease, hypertension, and atherosclerosis (*Conti et al., 2019*).

Our results showed a positive correlation between high AOPP levels and MPO activity in the IND, HFSD + IND, and OME + IND groups. The final product of MPO is hypochlorous acid (chlorinated oxidants) which reacts with plasma albumin to generate AOPP.

Pretreatment with *C. citrinus* and OME-ameliorated gastric ulceration damage induced by IND, *via* anti-inflammatory mechanisms, by decreasing MPO, COX-2, and 5-LOX activities and inflammatory cytokines (TNF-α and IL-6). These reductions suggest an antiulcerogenic ability, similar to the use of OME to reduce gastric damage. There are many reports of the protective effect of monoterpenes (carvacrol, limonene, 1-8-cineole, myrtenol, α-pinene, and α-terpineol) in models of gastric lesions (*Mansour et al., 2022*). However, studies have used high concentrations of these compounds. Consequently, plant products with these compounds in low concentrations and similar effects would be desirable as new gastroprotective alternatives.

*C. citrinus* extract contains limonene. This compound maintains normal levels of $PGE_2$ and reduces pro-inflammatory cytokines (*D'Alessio et al., 2013*), creating a protective effect on the epithelial barrier. *Wallace, Arfirs & McKnight (1991)* reported that ROS generated from neutrophils play a role in the vascular injuries caused by IND administration. Previously, we reported the strong antioxidant activity of *C. citrinus* extract (*Ortega-Pérez et al., 2022*). This antioxidant capacity and its effect on increasing COX-1 activity suggests the ability of *C. citrinus* to restore blood vessels.

Numerous studies have been reported about plants belonging to the Myrtaceae family showing protective effects against gastric ulcers. *Chanudom & Tangpong (2015)* showed that aqueous leaf powder extract of *Syzygium cumini* (L.) Skeels had a protective effect on IND-induced gastric ulcers in male mice, *via* the reduction of lipid peroxidation products and the enzymes involved in inflammation. *Thongsom, Chanudom & Tangpong (2019)*

showed that the administration of *Syzygium cumini* extract with 10 mg/kg of IND for 7 days reduced the ulcer index and some biomarkers of oxidative stress in male mice.

Methanol leaf extract of guava (*Psidium guajava*) had ulcer-protective effects on ethanol-induced gastric ulcers in adult nonpregnant female Wistar rats (*Uduak et al., 2012*). *de Almeida et al. (2017)* reported a total reduction of gastric lesions areas by using *Myrcianthes pungens* fruit and leaf extract administrated with 100 mg/kg of IND in female Swiss mice. Essential oil of *Melaleuca quinquenervia* inhibited GHS depletion and decreased MPO activity and MDA levels in a female Sprague-Dawley rat model of ethanol-induced peptic ulcers (*Cilingir-Kaya & Gurler, 2021*).

*Kar et al. (2021)* found that male and female Wistar rats administered aspirin (100 mg/kg) presented with the same reduction ulcer index and decreased gastric secretion after using methanol seed extract of *Syzyzium cumini*. *Keszei et al. (2010)* found that the Myrtaceae family had high terpenes concentration. Terpenes compounds have several biological antiviral, antioxidant, anti-inflammatory, immunomodulatory, and antiulcer activities (*Masyita et al., 2022*). Based on the previous review, the gastroprotective effects of *C. citrinus* should be related to the presence of terpenes that ameliorate the oxidative effects induced by IND. In future studies, it would be useful to examine the effect of the extract on both male and female animals with synchronized estrous cycles in order to eliminate any potential effects associated with female hormones.

Phenols and flavonoids (found in many fruits and vegetables) have a wide variety of pharmacological activities such as: anti-inflammatory, antiproliferative, antiangiogenic and anticancerogenic (*Serafim et al., 2020*). Despite the high concentration of phenols and flavonoids found in *Callistemon citrinus*, only quercetin, gallic acid and catechin have been reported to be used against gastric ulcers. Quercetin regulates apoptosis and COX and nitric oxide (NO) synthase activities in the ethanol-induced ulcer model in rats, and it also increases the antioxidant enzyme activities, nuclear translocation of the nuclear factor related to erythroid 2 (Nrf2), and prevents the factor kappa B (NF-κB) activation. Quercetin also generates the expression of P-selectin and intercellular adhesion molecule-1 (ICAM-1) in indomethacin-induced ulcer model in rats (*Alkushi & Elsawy, 2016*). Catechins present a gastroprotective effect *via* reduction in NO and iNOS levels and MPO activity (*Gil-Cardoso et al., 2017*) and also increase the activity of glutathione peroxidase, glutathione reductase and up-regulating of a Nrf2 and heme oxygenase-1 protein expression in Int-407 cells (*Cheng et al., 2013*). Because of the high content of phenolic compounds in *C. citrinus*, it would be desirable in a future study to analyze the role of main terpenes and phenolic compounds.

The results presented in this study showed that Wistar rats fed with a high fat-sugar diet and a supplement of *Callistemon citrinus* extract, at a daily dose of 250 mg/kg for 15 weeks, reduced the lesion of gastric ulcers and inflammation induced by indomethacin. However, the conclusions are limited to the gastroprotective effect of *Callistemon citrinus* in female rats. In future studies it would be desirable to evaluate the effect of the administration period and determine the therapeutic window.

## CONCLUSION

For the first time, this study showed that the administration of *C. citrinus* extract in Wistar rats fed with a HFSD prevented the risk of gastric ulcers after NSAID intake. *C. citrinus* extract reduced anti-inflammatory enzyme activities and pro-inflammatory cytokines. *C. citrinus* had a strong protective effect against gastric ulcers induced by IND.

## ACKNOWLEDGEMENTS

The authors thank the PhD Ma. Antonia Herrera-Vargas from the Instituto de Investigaciones sobre los Recursos Naturales de la UMSNH for the kind histopathological support.

### Funding

The authors received no funding for this work.

### Competing Interests

The authors declare that they have no competing interests.

### Author Contributions

- Jonathan Saúl Piñón-Simental performed the experiments, prepared figures and/or tables, and approved the final draft.
- Luis Alberto Ayala-Ruiz performed the experiments, prepared figures and/or tables, and approved the final draft.
- Luis Gerardo Ortega-Pérez analyzed the data, authored or reviewed drafts of the article, and approved the final draft.
- Oliver Rafid Magaña-Rodríguez performed the experiments, prepared figures and/or tables, and approved the final draft.
- Esperanza Meléndez-Herrera analyzed the data, authored or reviewed drafts of the article, and approved the final draft.
- Asdrubal Aguilera-Méndez analyzed the data, authored or reviewed drafts of the article, and approved the final draft.
- Patricia Rios-Chavez conceived and designed the experiments, analyzed the data, authored or reviewed drafts of the article, and approved the final draft.

### Animal Ethics

The following information was supplied relating to ethical approvals (*i.e.*, approving body and any reference numbers):

Institutional Committee for Use of Animals of the UMSNH (approval date: 01/12/2021; Protocol ID IIQB-CIBE-06-2021).

### Data Availability

The raw data is available at Zenodo: PIÑON SIMENTAL, J. S. (2023). Use of *Callistemon citrinus* as a gastroprotective and anti-inflammatory agent on indomethacin induced gastric ulcers in obese rats. In PEERJ. Zenodo. https://doi.org/10.5281/zenodo.10360356.

## Supplemental Information

Supplemental information for this article can be found online at http://dx.doi.org/10.7717/peerj.17062#supplemental-information.

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
