# Peer review of "Use of Callistemon citrinus as a gastroprotective and anti-inflammatory agent on indomethacin-induced gastric ulcers in obese rats"

_PeerJ, doi:10.7717/peerj.17062_

## Round 0.1 · original submission · Major Revisions

Dear authors, thank you for your submission. Please, refer to the reviewers comments for further details on the requested revisions and clarifications.

**Language Note:** The review process has identified that the English language must be improved. PeerJ can provide language editing services - please contact us at copyediting@peerj.com for pricing (be sure to provide your manuscript number and title). Alternatively, you should make your own arrangements to improve the language quality and provide details in your response letter. – PeerJ Staff

·

Basic reporting

-Professional English should be improved and should consult your colleague who is proficiency in English to review your manuscript.
-Miss typing, incomplete sentence should be approved.

Abstract :
-Your abstract needs more clear. I suggest that you should improve for more clearly understand of your study.
Keywords :
-“Obesity” is keyword in topic. The information about mechanism/pathological of gastric ulcer in obesity necessity have to literature review.

-Your introduction needs more detail. I suggest that you improve and clear the information about mechanism/pathological of gastric ulcer in obesity.

Experimental design

- The experimental design is not clear and needs to improve
You have mention in “In Biological material and preparation of 95% ethanol extract”. I suggest that you have to approve these all data in Table 1.
-The data of analysis should do by researcher. The % yield, gas chromatography-mass spectrometry (GC-MS) can show in the RESULTS. The extraction yield was 20% which is very high, is it right?
- The total phenols, flavonoids and terpenes should give the detail of analysis as well as the antioxidant capacity; DPPH, FRAP and ABTS, assays and present data in the RESULTS.
-You have to describe and briefly, explain how you did assays and the data should present in RESULTS.
-You aim to study the effect of C. citrinus on HFSD+indomethacin-induced gastric ulcer. However, the experimental designed group 4, 5, 6 female Wistar rats fed with food and water. Your experimental design was not clear and not response to hypothesis. Did Rat in group 4, 5, 6 fed with HFSD?
-Edit Line 195 to “3000rpm for 10min”
-Edit Line 199.” The final weight of groups 1, 2 and 3 was determined at the end of the experimental model.
-Line 218, 220. Please edit “the supertant”
-Line 220, should be “ Enzyme-linked immunosorbent assay (ELISA)”
-Line 262. Is Sulfuric acid?
- I suggest that you should briefly explain Statistical analysis.
- Figure 2 should be appropriated clearly present data.
-You should explain why only Table 3 that the data present are mean + SE (N = 6) difference from the other Figures and Tables.
-You should clearly explain more about Table 2. It is not appropriate and miss understand for data.

Validity of the findings

-You should clear that the female hormone of female mice that use in this study that will not interfere or effect on the treatment of C. I suggest that you give more explain in Discussion.
-Conclusions should be rewritten that should be well stated and linked to the advanced data discovery and research aim.

Additional comments

-This manuscript needs to be major improved.

Reviewer 2 ·

Basic reporting

This is an interesting manuscript where the authors explored the gastroprotective and
anti-inflammatory of Callistemon citrinus leaf extract

Experimental design

The authors used common methods and positive controls

Validity of the findings

The authors identified the anti-inflammatory and gastroprotective effects of a bioactive extract from Callistemon citrinus and annotated the individual components using GCMS. However, there are two major concerns that hinder the publication of the manuscript in its form.
1. This is a single-dose study; the authors should use at least three doses to validate the obtained results.
2. GCMS is not the appropriate analytical tool; the authors should annotate the chemical composition of the extract using LCMS.

---

## Round 0.2 · Minor Revisions

Dear authors, many thanks for your re-submission. Your manuscript still requires minor adjustments and clarifications before publication. (see attachment and reviewers comments). I ask for you to carefully proofread everything, and include some of the explanations to the reviewers in the manuscript particular the ones as significant as why a single dose study , where does that single dose come from and why. Also, the limitation in terms of phytochemical analysis, pros and cons of what you did / limitations. additionally, you should adapt the title to convey those key factors.

I am not sure about the relevance of Figure 1, because I cannot see the animals' weight evolution throughout the protocol. So this should be clear as well. average at the final is not that particularly significant if we don't know the average in the beginning, is it? don't forget that Latin names are always in italics!

The black arrows in Fig 2 should be more evident. you have arrows in Fig 3 that have not been mentioned in the legend. infiltration & hemorrhages should be clearly distinguished ie signaled. how was the histopathologic scoring done, by whom, for example?

All these details should be in the methods. So please, revise to make sure that all of your sections meet the highest patterns of scientific soundness and transparency. Although comprehensive, you need to carefully revise all the details and parameters to make sure that everything is being properly reported. Many thanks for your work.

·

Basic reporting

Title: -Please check and edit the title: Use of Callistemon citrinus as a gastroprotective and anti-inflammatory agent on indomethacin-induced gastric ulcers in obese rats
Abstract: Indomethacin is the keyword for this research investigation. Please give a mention/suggestion in the conclusion.
Introduction: Your introduction needs more details. I suggest that you improve the “ulcerogenic of high-fat diet/fatty acid should add more literature reviews. This will support your hypothesis of indomethacin and obesity-induced gastric ulcers. Moreover, the incidence of gastric ulcers in women, which is higher than men in obesity, needs to support the experimental design in this study.
Line 89: Please check and edit for NF-βB signaling.
English: The English language should be improved. Gamma needs to be checked and edited.

Experimental design

-Line 161-162: Please clarify for “fresh tissues” ? Is it fresh leaves?
- Line 163: Please clarify for the age of leaves “4-year-old leaves”. The standardization of C. citrinus extract was based on the use of 4-year-old leaves,….?
-Line 169-192. The induction of obesity and gastric ulcers experimental design has to clarify and improve.
-Line 245-249: Please clarify that Group 4, 5, and 6 were fed with HSSD or fed with rodent food (Rodent Diet®). And please give more detail of Three additional control groups (n = 6).
-Line 295, 306: Please add full name of MPO, COX-2
Line 295 add Cyclooxygenase (COX-1) and COX-2 activity
-Please check “µL, mL…..and make all consistency.
-Please improve the description at line 562-564.
Line 971: Please clarify “4-year-old leaves”
-Please provide statistical difference on bar graph “Values statistically deferent. Should it be * on bar graph and should show lower and upper error bar on graph not only upper bar)”

Validity of the findings

This article showed the impact and novelty assessed.

Additional comments

-This manuscript still needs to clearly written in professional.

Reviewer 2 ·

Basic reporting

The authors identified the anti-inflammatory and gastroprotective effects of a bioactive extract from Callistemon citrinus and annotated the individual components using GCMS.

Experimental design

The authors utilized common and well-established methods like GCMS, DPPH, FRAP, and in vivo model.

Validity of the findings

The authors highlighted the potential of Callistemon citrinus as an anti-inflammatory and gastroprotective agent.

Additional comments

Please highlight the shortcomings of the manuscript as a single dose and the role of the polyphenolics even though you have not identified them. you may refer to the literature.

---

## Round 0.3 · Minor Revisions

Dear authors, unfortunately, your manuscript still needs to go under careful proofreading before being accepted. Also, I advise you to re-check the guidelines for authors in terms of figures resolution. The methodology also needs to be carefully revised... not sure if you missed it last time? Please, refer to the reviewer's comments for further details.

·

Basic reporting

Abstract: Indomethacin is the keyword for this research investigation. The objective of this study should be; The aim of this study is to evaluate the gastroprotective effect of C. citrinus leaf extract on indomethacin-induced gastric ulcers in obese rats. It should relate to the title and research methods. Look at Lines 151-155 and 159-161 for ideas.
The abstract needs to be rewritten by a professional to improve the scientific style and clarity.
Introduction: Literature reviews were well done and updated.
English: The English language should be improved. by a professional to improve the scientific style and clarity.

Experimental design

Please list the chemicals and reagents used in this study and provide the source (company, city, country)
-Please re-writing Lines 247-251.
- Line 267: Please edit “clotted blood”
-Line 285: Please edit “µL” and check all.
-Line 287-289: Please check and edit.
-Line 374: p <0.05, a significant difference.?
-Based on Line 456, “IND produced high levels of MDA and HNE (lipid peroxidation products)” in RESULTS, then please provide the method/assay of MDA and HNE in METHODS. Also, check all results that should provide information on the assay method.
-Line 489-496: Please recheck and may need to improve.
-Line 534-537: Please check and need to improve.
-Line 544: PGE2 (Prostaglandin E2) and the other abbreviation words need the full names. Please check.

DISCUSSION and CONCLUSION need a professional edit.
Figures 2, 3, and 4 need to be presented clearly and with high resolution/magnification, and “P” consistency needs to be edited to match the others.

Validity of the findings

This article showed the impact and novelty assessed.

Additional comments

-This manuscript still needs to be clearly written professionally.

---

## Round 0.4 · accepted · Accept

Dear authors,

I want to express my appreciation for the effort you've put into this paper. It's evident that three of the co-authors, who are Titular Professors at the Universidad of Michoacan with over 20 years of research experience, bring invaluable expertise to any work. While acknowledging the esteemed credentials of all the co-authors, it's important to recognize that even seasoned researchers are susceptible to oversight. I deeply respect the diligent work and time our reviewers dedicate to reviewing manuscripts. It's possible that their feedback on language nuances extends beyond mere grammatical errors. It is also thanks to their diligence that forgetting to mention an assay in the methods while presenting data in the results section can be brought to attention.

On this occasion, I kindly request your attention to a couple of minor points for improvement. Firstly, please ensure consistency in formatting, such as using a different font size and style for the first paragraph and italicizing species names in the final version (abstract lines 39-43). Additionally, a small typo on line 57 requires correction—the article "the" is missing before "gastroprotective effect of...". Other small typos or mistakes have been also identified throughout (See attached). For future reference, it is well established not to start sentences with numbers, even when they are part of a chemical designation, such as "6-hydroxy-2,5,7,8-tetramethylchroman-2-carboxylic acid (Trolox)."

I want to assure you that there will be ample opportunity for thorough proofreading before publication. Addressing these minor issues will enhance the overall quality of your manuscript. Your cooperation in this matter is greatly appreciated.